# Fourteen-Year Temporal Trends in Patients Hospitalized for Mitral Regurgitation: The Increasing Burden of Mitral Valve Prolapse in Men

**DOI:** 10.3390/jcm11123289

**Published:** 2022-06-08

**Authors:** Clémence Grave, Christophe Tribouilloy, Philippe Tuppin, Alain Weill, Amélie Gabet, Yves Juillière, Alexandre Cinaud, Valérie Olié

**Affiliations:** 1Department of Non-Communicable Diseases, Santé Publique France, French Public Health Agency, 94410 Saint-Maurice, France; amelie.gabet@santepubliquefrance.fr (A.G.); valerie.olie@santepubliquefrance.fr (V.O.); 2Department of Cardiology, Amiens University Hospital, 80000 Amiens, France; tribouilloy.christophe@chu-amiens.fr; 3Department of Health Studies and Statistics, Caisse Nationale de l’Assurance Maladie, 75020 Paris, France; philippe.tuppin@assurance-maladie.fr (P.T.); alain.weill@assurance-maladie.fr (A.W.); 4Department of Cardiology, Nancy University Hospital, 54511 Vandoeuvre-lès-Nancy, France; yves.juilliere@wanadoo.fr; 5Department of Cardiology, Hôtel Dieu Hospital, 75004 Paris, France; alexandre.cinaud@aphp.fr

**Keywords:** epidemiology, hospitalization, heart valve diseases, mitral valve insufficiency

## Abstract

Mitral regurgitation (MR) is the second most common valvular heart disease in Europe. The aging of the population and the increase in post-infarction survival could increase the prevalence of MR. To estimate the burden of patients hospitalized for MR in France in 2019 and temporal trends by etiology and sex from 2006 to 2020, we selected all patients hospitalized for MR using the national hospital database. In 2019, 49.2% of such patients had mitral valve prolapse (MVP), 17.1% had ischemic MR, 9.9% had rheumatic MR and 4.4% had MR with cardiomyopathy. The mean age of MVP patients was 67.8 years, and 34% were women. Among 89% of MVP inpatients who had received mitral valve repair or replacement, 55% received surgical repair, 13% received percutaneous repair and 25% received replacement. The all-cause mortality of one year after a mitral procedure of MVP was 5.4%. Among ischemic MR inpatients, 29% have had a mitral valve replacement, 16% a surgical repair and 19% a percutaneous repair. Between 2006 and 2019, the age-standardized rates of patients hospitalized for MVP have increased by 60%, especially in men (+80%) with 5.3/100,000 Person-Years (PY). The age-standardized rates of patients hospitalized for ischemic MR have increased by 25% with 1.8/100,000 PY; that of rheumatic MR has decreased by 36%. The study found that the burden of MVP in hospitals has increased substantially, especially among men. These results emphasize the need to monitor these temporal trends and anticipate care needs in the coming years.

## 1. Introduction

Mitral regurgitation (MR) is the second most common valvular heart disease in Europe and the most common in the USA, with a prevalence of 9.3% in the population aged over 75 years [1,2,3,4]. Primary MR refers to abnormalities of the mitral valve leaflets and can remain asymptomatic despite underlying left ventricular enlargement and dysfunction. This ventricular damage leads to poor clinical outcomes with excess mortality, heart failure, and recurrence of atrial fibrillation. MR can also be functional with a normal mitral valve, which occurs due to annular dilatation and mainly left ventricular remodeling, leading to motion restriction in the normal mitral leaflets [5,6,7].

In Europe, 21% of native valvular heart diseases were MR, with one-third being primary MR in 2017 [4]. In developed countries, the main etiologies are degenerative, with valve prolapse for primary MR and ischemic for secondary MR [6,8]. The aging of the population can have a strong impact on the incidence of MR [9]. In addition, the epidemiology of the different cardiovascular diseases leading to MR is evolving and differs by sex. While the incidence of rheumatic heart fever has decreased [10], there is an increase in the incidence in of ischemic heart disease among young women [11] and an overall increase in post-infarction survival [12], which could increase the prevalence of MR. Finally, the management of MR has been improved by the advent of percutaneous techniques, enabling the management of patients with contraindications to other mitral procedures [13]. Thus, these advancements can influence the etiology and epidemiology of MR and its burden on the French hospital system. Moreover, due to differences in etiology and epidemiological factors of cardiovascular diseases and their management, sex differences could be observed.

The aim of this study was to estimate the burden of patients hospitalized for MR in a nationwide study in France in 2019 and to estimate temporal trends by etiology and sex from 2006 to 2020.

## 2. Materials and Methods

### 2.1. Data Sources and Population

This study was performed using the French National Health Data System (Système National des Données de Santé: SNDS), which provides detailed information on the real-life management of the entire population living in France (around 65,000,000 inhabitants) [14,15,16]. The SNDS is a medico-administrative tool comprising several databases linked by an individual anonymous number for each beneficiary enabling the follow-up of healthcare consumption. Among these databases, the French health insurance claim database contains demographic data and exhaustive data on all reimbursements for outpatient medical care: treatments, diagnostics, and therapeutic procedures. The SNDS also includes the National Hospital Discharge Database (Programme de médicalisation des systèmes d’information: PMSI), which records all hospitalizations in both public and private hospitals. For each hospital stay, the PMSI contains the principal diagnosis (PD), related diagnosis (RD), and associated diagnosis, coded according to the International Classification of Diseases-10 (ICD-10). Diagnoses are provided by a physician for each medical unit in which the patient is hospitalized and are determined at the end of their stay using medical records. PD and RD identify the reason for the hospital stay. The PD corresponds to the disease explaining the main reason of admission to the medical unit. The RD is an information that specifies when the PD is not informative enough, such as when the PD is an WHO-ICD10 chapter XXI code (“Z” code), and details the disease managed [17].

All patients hospitalized in France (excluding Mayotte, a French overseas territory) between 2006 and 2020, with a PD or RD of MR (ICD10 codes: I051 “Rheumatic mitral insufficiency”; I052 “Mitral stenosis with insufficiency”; I340 “Mitral insufficiency”; I341 “Mitral (valve) prolapse”), were selected. For each year of the study period, the first stay of the year when the patient was hospitalized for MR (i.e., PD or RD) was selected and defined as the index stay. This index stay was recorded to estimate the annual rate of patients hospitalized for MR.

The context of MR was studied to estimate the etiology while taking into account the patients’ comorbidities. Since 2006, the diseases were identified through inpatient diagnoses (PD, RD, and associated diagnosis) recorded during their hospital stays before the index stay. An algorithm for prioritizing etiologies was used to identify the most likely etiology of MR. Thus, a patient with a code for mitral prolapse (I341) or chordae tendineae rupture (I511) was categorized as mitral valve prolapse (MVP). In the absence of these codes, we searched for the code of acute rheumatic heart disease (I052; I051) followed by acute ischemic heart disease (I20–I25) and, finally, another cardiomyopathy code (I42; I43; I517; I518; I519; I52). Patients with codes of prolapse or rheumatic heart disease were classified as primary MR. Patients with MR in the context of chronic ischemic heart disease, cardiomyopathy or heart failure (I50; I110; I130; I132) or who underwent cardiac resynchronization therapy with a triple-chamber cardioverter-defibrillator or pacemaker procedure were classified as a secondary MR. Patients with none of these codes were categorized as “unclassified MR” (Appendix A). The history of heart failure was identified by hospitalization diagnoses for hospital stays between 2006 and index stay (I50; I110; I130; I132)

All hospital readmissions and readmissions for MR and heart failure were identified within the year following the index stay or the mitral procedure. Major adverse cerebral and cardiovascular events (MACCE) (all-cause deaths, stroke, acute coronary syndrome, systemic embolism, cardiogenic shock, MR or heart failure) were also identified during 1-year follow-up after surgical or percutaneous act.

The PMSI database also contains information about specific procedures performed during the hospital stay. Mitral valve procedures—i.e., mitral valve replacement (DBKA002; DBKA005; DBKA009; DBKA010; DBMA005; DBMA013), mitral valve plastic surgery (DBMA002; DBMA003; DBMA007) and percutaneous mitral valve repair (DBBF198)—coded using the French classification of medical procedures (Classification Commune des Actes Médicaux: CCAM) were searched during the index hospital stay and all subsequent hospital stays in the following rolling year. The CCAM code DBBF198, corresponding to percutaneous mitral valve repair, was introduced into this classification in December 2016.

Several sociodemographic and medical characteristics (age, sex, number of hospital stays, length of stay, MR management, heart failure, all-cause mortality) were extracted directly or reconstructed (geographic social deprivation, Charlson comorbidity index) from the data available in the SNDS database. The Charlson comorbidity index was calculated using the method of Quan et al. [18]. The level of social deprivation was estimated from the French deprivation index (FDep) developed by Rey et al. [19]. The FDep is an ecological indicator that defines population quintiles according to the level of social deprivation in the municipality of residence (the smallest administrative units in France). For our analyses, we used the FDep by dividing the general population into quintiles and calculated the data for the most recent year available (2013). The scale is based on the place of residence according to four factors: mean household income, percentage of high school graduates among inhabitants aged 15 years and older, percentage of manual workers in the working population, and unemployment rate. The first quintile (Q1) group represents the least disadvantaged group and the fifth quintile (Q5) the most disadvantaged group in France.

### 2.2. Statistical Analysis

Sociodemographic and medical characteristics of patients hospitalized for MR were described for 2019 (the most recent available year, before the Covid-19 pandemic) and were classified by sex and MR etiology. Qualitative variables were compared using Chi2 test (or Fisher exact test) and quantitative variables using *t*-test. We performed logistic regressions to estimate the sex-differences by mitral procedure types (adjusted by age and Charlson index) and mortality rates (adjusted by age, Charlson index and mitral procedure types). A subgroup analysis of patients with a previous hospitalization for heart failure was performed. Crude and age-standardized rates (standardized to the 2010 European population) of hospitalized MR patients were computed for the overall population and then separately by sex and MR etiology. The rate of patients hospitalized for MR was defined as the number of patients hospitalized for MR at least once during the year divided by the number of French inhabitants. This rate was expressed per 100,000 Person-Years (PY). The national average population data for 2006 to 2020 was taken from the National Institute of Statistics and Economic Studies (INSEE). Absolute and relative differences in rates were computed on data from 2006–2019, not including 2020, to assess overall trends prior to the COVID-19 pandemic. Poisson regression was applied to calculate the average annual percent change between 2006 and 2019 by age group and sex using population log as an offset variable.

Statistical analyses were performed with the SAS software (version 7.11, SAS Institute Inc., Cary, NC, USA).

## 3. Results

### 3.1. Characteristics of Patients Hospitalized for MR in 2019, According to Etiology

In 2019, 7232 patients were hospitalized for MR, and 89% of the cases were estimated for the etiology of MR (Appendix A). Table 1 and Table 2 described the characteristics of primary and secondary MR patients by etiology and sex.

#### 3.1.1. Primary MR

In 2019, 3560 patients were hospitalized for MVP, and 34.3% were women. The mean age of these patients was 67.8 years, being higher in women than in men (71.8 years vs. 65.8 years) (*p* < 0.001). The mean Charlson comorbidity index score was similar for MVP among men and women, although women were more likely to be hospitalized for heart failure. Concerning patient management, 55.1% had a surgical mitral valve repair (MVR), 14.7% a percutaneous MVR and 19.0% a surgical valve replacement, while 11.2% were not treated during the index hospital stay or the following year. Regarding the between-sex differences, 14.3% of women did not undergo a procedure as compared to 9.6% of men (*p* < 0.0001). This sex-difference remained significant after the adjustment of age and comorbidities. Of MVP, 19.8% of women (mean age = 84 years) and 12.0% of men (mean age = 79 years) benefited from percutaneous MVR. After adjustment for age and comorbidities, there was no difference for percutaneous MVR (OR_adj_ = 0.94 [0.75;1.18] *p* = 0.61, Appendix A), but women received less surgery for MVP (OR_adj_ = 0.70 [0.60;0.82], *p* < 0.01, Appendix A). Women hospitalized for MVP had higher rates of MACCE in the year following the surgical or percutaneous act (14.4% in women vs. 10.5% in men, *p* = 0.001). In the year following the index stay, the all-cause mortality rate was 8.1% in women (mean age = 81 years) and 5.4% in men (mean age = 76 years) (*p* = 0.002). After controlling for age, comorbidities and management, the difference was not significant (1.05 [0.78;1.41], *p* = 0.77). Age and comorbidities were strongly associated with mortality (Appendix A).

The subgroup of patients with MVP and previous hospitalization for heart failure had more comorbidities and were older than those patients without a history of heart failure. They benefited more from percutaneous MVR (29.1% vs. 8.2%) and surgical valve replacement (24.4% vs. 16.6%) (Appendix A).

The hospitalization for rheumatic MR involved 717 patients in 2019, with 69.2% being women and a mean age of 65.1 years. In total, 55.2% had a surgical valve replacement, 9.9% a surgical MVR and 4.6% a percutaneous MVR, while nearly one-third of rheumatic MR patients were not treated during the index hospital stay or the following year. The proportion of surgical valve replacement was higher among women than men (58.3% vs. 48.4%–OR_adj_ = 1.53 [1.11;2.11], *p* = 0.02). Patients with rheumatic MR had a high comorbidity index, with an all-cause mortality rate at 1 year of 10.6% after their index stay (mean age = 75 years). Those who underwent surgery or a percutaneous procedure had an all-cause mortality rate of 9.0% at 1 year (mean age = 71 years) (Table 2).

#### 3.1.2. Secondary MR

In 2019, 2127 patients were hospitalized for secondary MR; 43.3% were women and the mean age of these patients was 72.1 years (men: 69.7; women: 72.2). Among 2127 patients, 1235 were hospitalized for ischemic MR (58.1% of secondary MR, 36.0% women), 319 for cardiomyopathy MR (15.0% secondary MR, 53.0% women) and 573 for other secondary MR (26.9% of secondary MR, 53.4% women). A socioeconomic gradient was observed for secondary MR, with a higher proportion of MR patients in the most socially disadvantaged quintiles (Q5) than in the least disadvantaged quintiles (Q1). During the index stay or in the following year, 28.2% of secondary MR patients underwent a surgical valve replacement, 19.0% a mitral valve repair, and 16.3% a percutaneous MVR. A high proportion of secondary MR patients did not undergo a mitral procedure (42.9% of cardiomyopathy MR, 35.3% of ischemic MR), especially women (Table 2). This difference in the management remained significant after the adjustment for age and comorbidities (Appendix A). Among patients who underwent a mitral valve procedure, 23.4% were readmitted for MACCE within 1 year. The all-cause mortality rate after 1 year of hospitalization was 15.4% for ischemic MR, 15.0% for cardiomyopathy and 12.2% for other secondary MR.

Unclassified MR patients were younger, less comorbid and their death rates were lower (Appendix A).

### 3.2. Temporal Trends

In 2019, the age-standardized rate for patients hospitalized for MR was 10.8 per 100,000 PY (Appendix A and Appendix A). The proportion of MR on MVP had increased since 2006. The proportion of rheumatic and unclassified MR had decreased (Appendix A).

#### 3.2.1. Primary MR

Between 2006 and 2019, the age-standardized rates of patients hospitalized for MVP had an increased rate of 5.3/100,000 PY in 2019 (+80%), with a higher rate for men (7.7/100,000 PY, +89%) than for women (3.2/100,000 PY, +60%) (Figure 1). This increase was significant among all sexes and age subgroups older than 45 years (Figure 2a). In 2020, there was a decrease in age-standardized rates for patients hospitalized for MVP (4.8/100,000 PY).

The age-standardized rate of patients hospitalized for rheumatic MR had decreased over the same period, from 1.7/100,000 PY in 2006 to 1.1/100,000 PY in 2019 and to 0.8/100,000 PY in 2020 (Figure 1). This rate decreased in both sexes among less than 85 years of age (Figure 2b). The incidence remained higher in women (1.4/100,000) than in men (0.7/100,000 PY).

#### 3.2.2. Secondary MR

Between 2006 and 2019, the age-standardized rates of patients hospitalized for ischemic MR had increased by 26%, reaching a rate of 1.8/100,000 PY in 2019, (+16.3%—Figure 3). The annual percentage change in rates was significantly increased in the oldest age groups and among women aged 55 to 64 (Figure 4).

The age-standardized rates of patients hospitalized for cardiomyopathy with MR and other secondary MR were stable from 2006 to 2019 and were 0.48/100,000 PY and 0.84/100,000 PY, respectively (Figure 3).

### 3.3. Figures, Tables and Schemes

The interpretation should take into account the decrease in the proportion of unclassified MR between 2006 and 2020.

## 4. Discussion

Recent 14-year trends showed an increase in the annual rate of patients hospitalized for MR in France, especially among men and primary MR. This study described the temporal trends, and these results can be used to monitor MR hospitalization and changes in profile, management and vital prognosis of MR inpatients.

This study confirms the expected trends and quantified the burden of each etiology in the epidemiology of MR. The increase in patients hospitalized for MR could be due to several phenomena: the increase of MVP due to ageing of population and the increase of HF and secondary MR.

The majority of patients hospitalized for MR, had a primary MR mainly due to mitral valve prolapse [3,8,9,20]. The proportion of hospitalized patients with MVP has increased significantly over the last 14 years in line with the ageing population. On the contrary, rheumatic etiology concerned less than 10% of patients hospitalized for MR in 2019, showing a decrease since 2006 in accordance with incidence trends of rheumatic fever in developed countries [10]. Secondary MR is also a challenge as it accounted for one-third of hospitalizations related to the MR in 2019. Ischemic MR is the most common etiology for secondary MR. As expected, we found that secondary MR patients had different characteristics and prognosis in comparison to primary patients.

This study highlighted the epidemiological differences between men and women among both primary and secondary MR inpatients. Part of the sex differences in inpatients rates can be explained by the epidemiology of the etiologies. Firstly, the incidence of ischemic heart disease was higher among men, but this difference decreased with increased incidence in young women [12,21,22]. This might be explained by there being more men hospitalized for secondary MR, but this rate was increased more among women. Secondly, as compared to MR prevalence studies [1,9], this study found a lower rate of hospitalization for MVP in women than in men. This could be explained due to the under-diagnosis and/or under-treatment of women with degenerative MR [5,23]. The higher proportion of history of heart failure among the older-aged women hospitalized for MR could suggest that women were hospitalized at a more advanced stage of their illness, a longer survival of women with MR or a later presentation of MR.

The patients selected in our study were those who were hospitalized for MR, for the treatment or MR assessment to evaluate the need and possibility of valve intervention. This explained why a large proportion of patients included in the study were undergoing mitral valve surgery (surgical or percutaneous) during the index stay or during a readmission for MR in the following year. Hospitalization for MR assessment could explain the high rates of rehospitalisation for MR and more than 10% of patients hospitalised for MR did not undergo mitral procedure in the following year of the index stay. Our epidemiological study and database could not provide more precise information regarding these patients. In particular, we could not know whether the patient had an indication or contraindication for one or all of the mitral procedures, and we did not have the preoperative evaluation (such as the EuroSCORE II). The intervention indications and severity of MR could not be evaluated in this study. Nevertheless, these results were adjusted for age and comorbidities and suggested the sex differences in MR management and on mitral repair. These results should be interpreted with caution and should be complemented by studies with indication and more clinical information [24]. The French study by Messika-Zeitoun et al. highlighted that repair rates were suboptimal for all MR [20].

In addition, the MR management data highlighted the development of percutaneous mitral repair techniques and this identification in our medico-administrative database was only possible since December 2016. Nevertheless, our study had already showed the increased use of this technique. Despite the limited European Society of Cardiology (ESC) 2017 recommendations on percutaneous management [13], we found that 12% of men and nearly 20% of women with MVP underwent a percutaneous valve procedure during their index stay or within a year. These patients had a mean age of 81 years, which appeared to be in accordance with the guidelines. Similarly, MVP patients with associated heart failure had more percutaneous mitral procedures. Secondary MR was associated with poor prognosis, and the efficacy of surgical treatment and percutaneous MVR is still being discussed [13,25,26,27,28,29,30,31,32,33,34].

This real-life, nationwide study assessed the all-cause mortality at one year after hospitalization or mitral procedure, including out-of-hospital mortality. The mean age of patients who died in the following year of MR hospitalization was 76.3 years for men and 81.4 years for women for MVP and ranged from 73 to 77 years for other MR. By comparing the age of death with the life expectancy in 2019 in France, it was 79.8 years for men and 85.6 years for women [35,36]. In addition to the frequent comorbidities of secondary MR patients were compared to the overall population of the same age. The direct effect of MR that leads to left ventricular dilatation and left ventricular ejection fraction alteration can begin as early as the middle MR stage during the asymptomatic phase. Nevertheless, even with a preserved left ventricular ejection fraction, excess mortality has been described in MR patients in terms of both primary and secondary MR (higher for secondary than for primary MR) [5]. The long-term survival of patients with chronic ischemic MR was worse than that of MR patients with other etiologies [37]. We found the highest mortality rates for patients hospitalized for ischemic MR, reaching 17.5% in women. Differences in patient age and clinical condition (comorbidities) during the index stay differed between the sex and etiology subgroups and explained part of the observed differences in prognosis.

Globally, this study provided information on mortality and readmission following a hospitalization for MR. The differences in patients’ characteristics were observed according to the etiology of the MR and the sex. Adjusted analyses were performed to partially explain the observed differences on mortality, but some factors could not be taken into account, such as the severity of MR.

In early 2020, the COVID-19 pandemic significantly affected the French healthcare system, including both in and out-of-hospital care, which led to the cancelling of scheduled non-emergency care on one hand and the saturation of some services, especially those linked to COVID-19 management, on the other hand. As observed for several cardiovascular diseases [38,39], our study found that in 2020, there was a decrease in hospital admissions for MR. A decrease in several cardiac procedures was highlighted in England in early 2020, and mitral procedures were among the most impacted [40]. During a time of major restrictions in services and during peak of pandemic, mitral procedures were recommended only in emergencies or in the most severe patients not controlled by drug therapy [41]. Ambulatory and telemedicine management was encouraged [42] and could explain part of the decrease in the MR hospital admission in 2020. In addition, some patients were able to delay their consultations or follow-ups on their own, for fear of infection or due to restrictions imposed during the lockdown.

Another hypothesis that could explain the 2020 decrease observed in our study could be the overall decrease in healthcare in France, particularly in general practitioner and cardiologist visits, that may have limited the detection of MR and led to a decrease in their management. These hypotheses highlight the need for ongoing monitoring of this disease because delays in management could have an impact on the epidemiology of hospitalizations for MR in the coming years, possibly leading to an increase in the number of mitral valve repair or replacement or MR complications. Finally, patients with comorbidities or a history of cardiovascular disease were at particular risk of dying from COVID-19 in 2020. Thus, we cannot completely rule out that COVID-19-related mortality may have been an important competitive cause of death and could have impacted the number of patients with MR [43].

### Strengths and Limitations

The French National Health Data System is a comprehensive database covering the entire French population [15]. Our study was able to estimate the annual rate of patients hospitalized for MR accurately and provided real-life observations on mitral procedure rates and all-cause mortality rates. The patients were selected on the PD, and RD MR was the reason for hospital stay and did not consider the less significant MR (associated diagnosis). The extrapolation of these indicators should be analyzed with caution, since our hospital indicator depends on the MR incidence rate but also on the diagnosis, severity (not available on our medico administrative database) and hospital management of MR.

The study by Messika-Zeitoun et al. [44] estimating the societal burden of MR in France showed a higher number of patients hospitalized in France. The selection method of this study was based on primary and secondary diagnoses and reflected more prevalence of the disease in France. Our study included only those patients who were hospitalized with a primary diagnosis of MR and provided an estimation of the annual incidence of patients hospitalized for MR. Due to the different populations selected, the characteristics of the patients (age, mortality rate especially) could differ. Nevertheless, the distribution of MR causes (primary vs. secondary MR) were similar, and the subpopulation with valve intervention was close to our data. Through this global vision, our indicator showed the evolution of the incidence of MR, the use of hospitalization and MR management. Thus, these indicators are useful for monitoring the temporal trends in real time, estimating the burden of MR, and anticipating the hospital’s healthcare needs.

The study had several limitations inherent to medical administrative databases. Clinical information such as MR severity was not available. We only had information on the existence of concomitant heart failure. A misclassification bias related to the coding of diagnoses (MR and etiology) in hospitals cannot be excluded. The decrease in the number of unclassified MR indicates that the accuracy of our etiology classification algorithm is improving over time, but this may impact the evolution of classified MR. However, the changes in etiological distribution seem to be in line with the literature and physio-pathological hypotheses (decrease in acute rheumatic fever, ageing population, increase in heart failure), and the descriptive profile in each group is consistent with the epidemiology of these different diseases and the guidelines for the management of MR [4,13,20,23]. Finally, only all-cause mortality was available in our database, which did not allow us to identify the cardiovascular mortality, which led us to include all-cause mortality in the MACCE outcome.

## 5. Conclusions

The burden of MVP in hospitals has increased substantially, especially among men. The secondary MR hospitalization rate has increased, especially among women and ischemic MR. These observations might be related to a shift in context of both the etiology and management of MR. These results emphasized the need to monitor these temporal trends and anticipate care needs in the coming years and to detect differences in the management and the mortality among men and women.

## Figures and Tables

**Figure 1 jcm-11-03289-f001:**
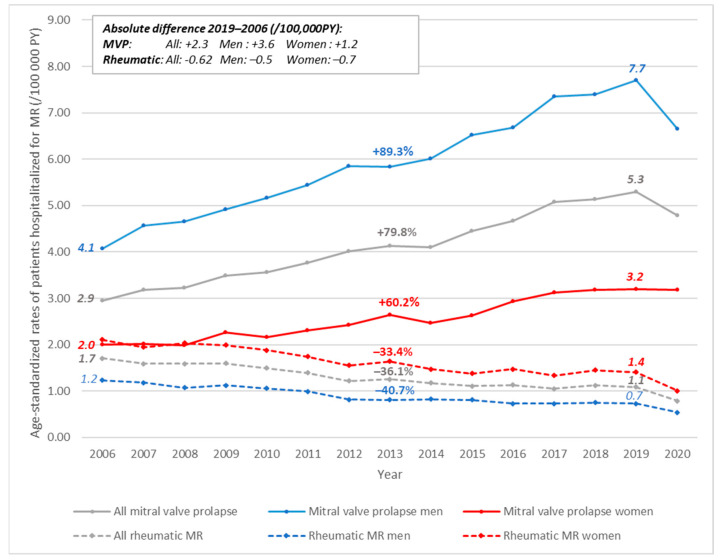
Trends in age-standardized rates of patients hospitalized for primary mitral regurgitation in person-years, according to etiology and sex, France, 2006–2020. *The percentage above the curve is the relative rate difference between 2019 and 2006. PY = person-years; MVP = mitral valve prolapse*.

**Figure 2 jcm-11-03289-f002:**
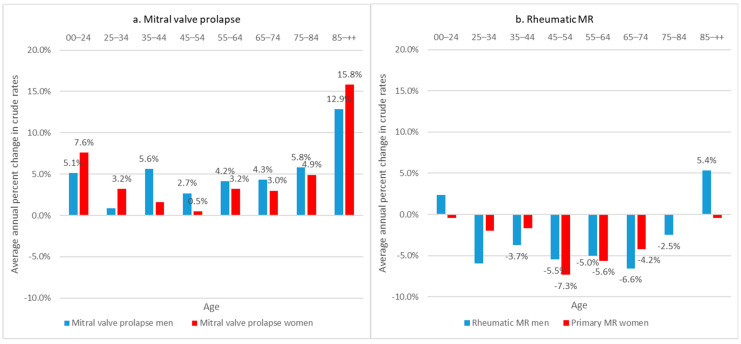
Average annual percent change in crude rates of patients hospitalized for primary mitral regurgitation, according to etiology and sex, France, 2006–2019. *Only values significantly different from 0% are presented (alpha risk = 5%)*.

**Figure 3 jcm-11-03289-f003:**
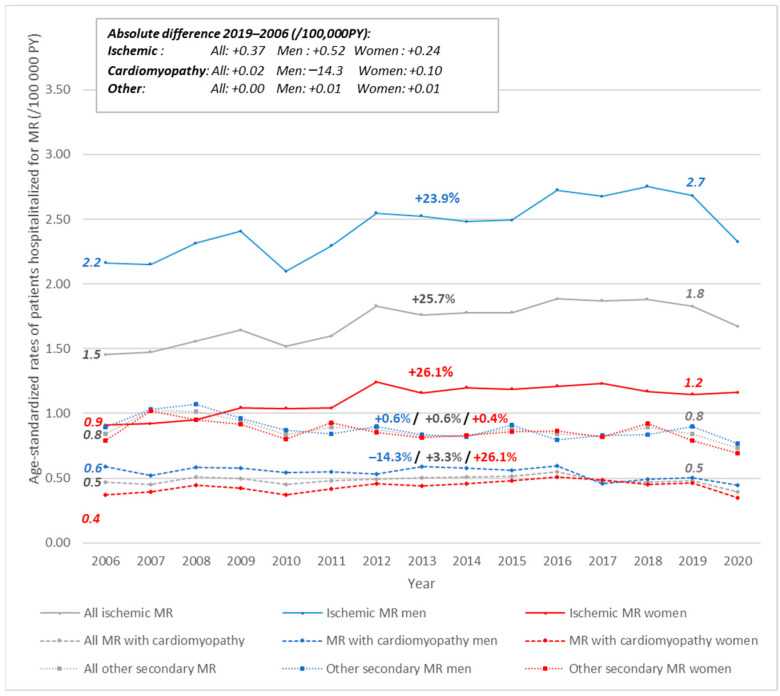
Trends in age-standardized rates of patients hospitalized for secondary mitral regurgitation in person-years, according to etiology and sex, France, 2006–2020. *The percentage above the curve is the relative rate difference between 2019 and 2006. PY = person-years*.

**Figure 4 jcm-11-03289-f004:**
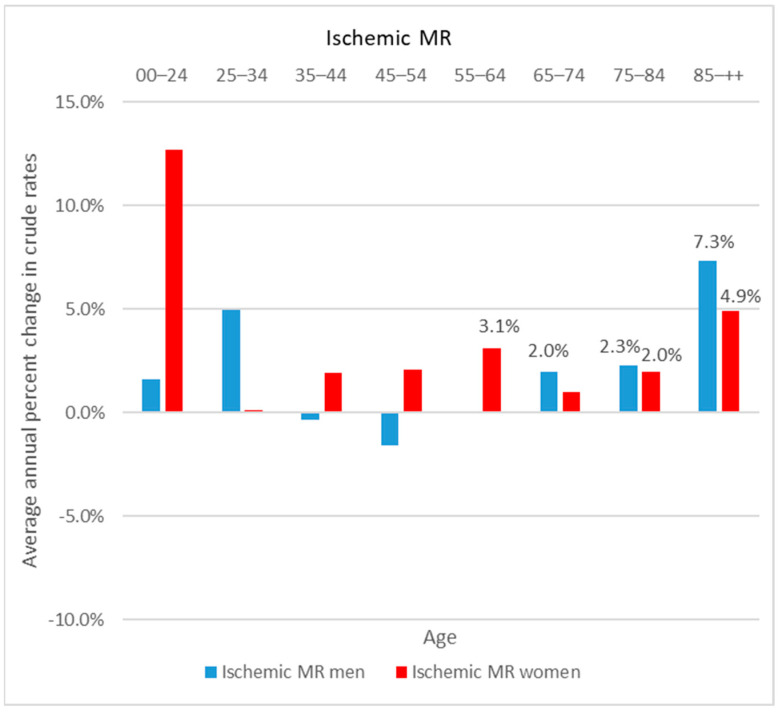
Average annual percent change in crude rates of patients hospitalized for ischemic mitral regurgitation, according to sex, France, 2006–2019. *Only values significantly different from 0% are presented (alpha risk = 5%)*.

**Table 1 jcm-11-03289-t001:** Characteristics of patients hospitalized for primary mitral regurgitation according to etiology and sex, France, 2019.

2019	Primary RM	Mitral Valve Prolapse				Rheumatic RM			
		All	Men	Women	*p*-Value	All	Men	Women	*p*-Value
**Total number, n**	4277	3560	2338	1222		717	221	496	
**Demographic characteristics**	-								
Age (years), mean (SD)	67.4 (14.0)	67.8 (13.5)	65.8 (13.0)	71.8 (13.7)	<0.0001	65.1 (15.9)	64.0 (16.8)	65.5 (15.5)	0.22
**Fdep *, n (%)**									
Quintile 1 (the least disadvantaged)	832 (20.2%)	20.0%	20.6%	18.9%	0.41	21.1%	25.0%	19.3%	0.56
Quintile 2	827 (20.1%)	20.0%	20.4%	19.4%		20.2%	18.0%	21.1%	
Quintile 3	897 (21.8%)	22.0%	21.2%	23.7%		20.3%	19.5%	20.7%	
Quintile 4	807 (19.6%)	19.8%	20.0%	19.6%		18.2%	18.0%	18.2%	
Quintile 5 (the most disadvantaged)	760 (18.4%)	18.1%	17.9%	18.4%		20.3%	19.5%	20.7%	
**Medical characteristics**									
Charlson comorbidity index mean (SD)	1.96 (2.11)	1.9 (2.1)	1.9 (2.1)	1.9 (2.0)	0.86	2.5 (2.2)	2.7 (2.4)	2.4 (2.1)	0.05
History of hospitalization for heart failure	1447 (33.8%)	30.8%	28.1%	36.0%	<0.001	48.7%	47.5%	49.2%	0.68
Length of index stay mean (SD)	9.1 (10.8)	9.0 (10.5)	9.0 (8.6)	8.9 (8.3)	0.65	9.9 (11.9)	10.2 (13.4)	9.7 (11.2)	0.61
**Management of mitral regurgitation in the year following the index hospital stay **, n (%)**									
Surgical valve replacement	1073 (25.1%)	19.0%	18.0%	20.9%	0.03	55.2%	48.4%	58.3%	0.01
*Mean age*	*67.1*	*69.4*	*68.4*	*71.1*		*63.2*	*60.3*	*64.3*	
Surgical valve plasty	2032 (47.5%)	55.1%	60.4%	44.9%	<0.001	9.9%	15.8%	7.3%	<0.001
*Mean age*	*62.8*	*63.0*	*61.5*	*66.8*		*58.1*	*62.8*	*53.6*	
Percutaneous mitral valve repair	555 (13.0)	14.7%	12.0%	19.8%	<0.001	4.6%	8.6%	2.8%	<0.001
*Mean age*	*81.2*	*81.4*	*79.3*	*83.9*		*77.0*	*77.3*	*76.6*	
No mitral procedure mentioned	617 (14.4%)	11.2%	9.6%	14.3%	<0.001	30.3%	27.1%	31.7%	0.23
*Mean age*	*70.5*	*71.4*	*70.8*	*72.1*		*68.8*	*67.0*	*69.5*	
**Readmission in the year following**									
After index stay, all causes, n (%)	2600 (60.8%)	59.9%	59.5%	60.9%	0.41	65.0%	70.1%	62.7%	0.05
After index stay, for RM, n (%)	1521 (35.6%)	37.1%	36.2%	38.8%	0.13	28.0%	34.8%	25.0%	0.007
After index stay, for heart failure, n (%)	327 (7.6%)	6.6%	5.6%	8.4%	0.001	13.0%	11.3%	13.7%	0.38
After surgical or percutaneous act, all causes, n (%)	1284 (35.1%)	33.6%	33.9%	33.0%	0.58	44.4%	45.3%	44.0%	0.77
After surgical or percutaneous act, for RM, n (%)	80 (2.2%)	2.3%	1.9%	3.2%	0.03	1.0%	3.1%	0.6%	0.04
After surgical or percutaneous act, for heart failure, n (%)	196 (5.4%)	4.7%	4.4%	5.4%	0.17	9.4%	7.5%	10.3%	0.30
After surgical or percutaneous act, MACCE, n (%)	469 (12.8%)	11.8%	10.5%	14.4%	0.001	19.2%	18.6%	19.5%	0.82
**All causes mortality, n (%)**									
**After index stay**									
During the index hospital stay	65 (1.5%)	1.4%	1.0%	2.1%	0.005	2.2%	2.7%	2.0%	0.56
*Mean age*	*74.2*	*76.4*	*73.7*	*78.8*		*67.4*	*67.2*	*67.6*	
At 30 days	109 (2.5%)	2.4%	1.8%	3.5%	0.002	3.2%	2.7%	3.4%	0.62
*Mean age*	*76.1*	*77.9*	*75.0*	*80.8*		*69.6*	*67.2*	*70.5*	
At 1 year	302 (7.1%)	6.3%	5.4%	8.1%	0.002	10.6%	11.3%	10.3%	0.68
*Mean age*	*77.6*	*78.5*	*76.3*	*81.4*		*74.7*	*76.3*	*73.9*	
**After surgical or percutaneous act**									
At 30 days	86 (2.3%)	2.1%	1.8%	2.8%	0.06	4.0%	2.5%	4.7%	0.23
*Mean age*	*74.5*	*75.6*	*72* *.7*	*79.4*		*70.7*	*69.8*	*70.9*	
At 1 year	197 (5.4%)	4.8%	4.0%	6.4%	0.003	9.0%	9.3%	8.8%	0.86
*Mean age*	*76.3*	*77.8*	*75.7*	*80.4*		*71.4*	*72.5*	*70.8*	

* available only for metropolitan France; ** Index hospital stay = the first stay of the year when the patient was hospitalized for MR.

**Table 2 jcm-11-03289-t002:** Characteristics of patients hospitalized for secondary mitral regurgitation according to etiology and sex, France, 2019.

2019	Secondary RM	Chronic Ischemic Heart Disease MR				MR with Cardiomyopathy				Other Secondary MR			
		All	Men	Women	*p*-Value	All	Men	Women	*p*-Value	All	Men	Women	*p*-Value
**Total number, n**	2127	1235	790	445		319	150	169		573	267	306	
**Demographic characteristics**													
Age (years), mean (SD)	72.1 (12.3)	72.9 (10.7)	71.9 (10.4)	74.8 (10.9)	<0.001	68.5 (14.3)	67.3 (14.2)	69.5 (14.4)	0.17	72.3 (13.8)	69.8 (14.2)	74.4 (13.0)	<0.001
**Fdep *, n (%)**													
Quintile 1 (the least disadvantaged)	358 (17.5%)	17.5%	18.7%	15.5%	0.54	17.4%	20.8%	14.4%	0.07	17.6%	16.5%	18.5%	0.19
Quintile 2	406 (19.9%)	19.3%	18.5%	20.7%		22.4%	23.6%	21.3%		19.8%	20.1%	19.5%	
Quintile 3	431 (21.1%)	22.1%	22.7%	20.9%		18.8%	13.2%	23.8%		20.3%	23.6%	17.5%	
Quintile 4	421 (20.6%)	21.1%	20.5%	22.1%		18.4%	21.5%	15.6%		20.9%	22.0%	19.9%	
Quintile 5 (the most disadvantaged)	425 (20.8%)	20.0%	19.6%	20.7%		23.0%	20.8%	25.0%		21.4%	17.7%	24.6%	
**Medical characteristics**													
Charlson comorbidity index mean (SD)	3.1 (2.0)	3.2 (2.2)	3.1 (2.1)	3.2 (2.2)	0.51	2.9 (1.9)	2.8 (1.9)	2.9 (1.9)	0.55	3.1 (1.7)	3.0 (2.8)	3.1 (2.9)	0.65
History of hospitalization for heart failure	1371 (64.5%)	63.4%	61.8%	66.3%	0.11	68.3%	64.0%	72.2%	0.12	64.6%	60.3%	68.3%	0.05
Length of index stay mean (SD)	10.2 (12.4)	10.5 (13.4)	10.8 (13.7)	10.0 (12.7)	0.30	9.2 (11.0)	9.0 (11.6)	9.3 (10.5)	0.80	9.9 (11.1)	10.0 (9.6)	9.9 (12.3)	0.92
**Management of mitral regurgitation in the year following the index hospital stay **, n (%)**													
Surgical valve replacement	600 (28.2%)	29.1%	28.7%	29.7%	0.73	26.7%	22.7%	30.2%	0.13	27.2%	26.2%	28.1%	0.61
*Mean age (years)*	*69.0*	*69.7*	*69.3*	*70.5*		*66.4*	*62.6*	*68.9*		*68.7*	*68.1*	*69.2*	
Surgical valve plasty	405 (19.0%)	16.2%	19.2%	10.8%	<0.001	18.8%	25.3%	13.0%	<0.001	25.3%	32.2%	19.3%	<0.001
*Mean age (years)*	*66* *.7*	*68.3*	*67.5*	*70.6*		*62.6*	*63.1*	*61.9*		*66.1*	*65.2*	*67.5*	
Percutaneous mitral valve repair	347 (16.3%)	19.4%	20.5%	17.5%	0.20	11.6%	16.0%	7.7%	0.02	12.2%	13.9%	10.8%	0.26
*Mean age (years)*	*77.8*	*77.2*	*75.8*	*80.1*		*76.0*	*75.8*	*76.5*		*80.9*	*80* *.4*	*81* *.4*	
No mitral procedure mentioned	775 (36.4%)	35.3%	31.5%	42.0%	<0.001	42.9%	36.0%	49.1%	0.02	35.3%	27.7%	41.8%	<0.001
*Mean age (years)*	*74.8*	*75.4*	*74.4*	*76.7*		*70.3*	*69.6*	*70.7*		*76.4*	*71.6*	*79.2*	
**Readmission in the year following**													
After index stay, all causes, n (%)	1277 (60.0%)	62.1%	60.5%	64.9%	0.12	62.4%	68.0%	57.4%	0.05	54.3%	55.1%	53.6%	0.73
After index stay, for RM, n (%)	426 (20.0%)	18.9%	18.5%	19.8%	0.58	20.1%	22.7%	17.8%	0.27	22.3%	24.7%	20.3%	0.20
After index stay, for heart failure, n (%)	318 (15.0%)	16.6%	15.1%	19.3%	0.05	12.2%	10.7%	13.6%	0.42	12.9%	14.6%	11.4%	0.26
After surgical or percutaneous act, all causes, n (%)	582 (43.0%)	46.3%	44.9%	49.2%	0.25	43.4%	49.0%	37.2%	0.11	35.8%	35.8%	36.0%	0.97
After surgical or percutaneous act, for RM, n (%)	24 (1.8%)	1.6%	1.5%	1.9%	0.77	1.6%	1.0%	2.3%	0.60	2.2%	2.1%	2.2%	0.91
After surgical or percutaneous act, for heart failure, n (%)	135 (10.0%)	11.3%	10.0%	14.0%	0.10	9.3%	9.4%	9.3%	0.99	7.6%	8.8%	6.2%	0.34
After surgical or percutaneous act, MACCE, n (%)	316 (23.4%)	24.4%	21.4%	30.6%	0.005	24.7%	20.8%	29.1%	0.20	20.5%	17.1%	24.2%	0.09
**All causes mortality, n (%)**													
**After index stay**													
During the index hospital stay	89 (4.2%)	4.5%	3.7%	5.8%	0.08	4.1%	2.7%	5.3%	0.23	3.7%	3.0%	4.2%	0.43
*Mean age (years)*	*75.2*	*75.1*	*74.5*	*75.7*		*75.5*	*67.3*	*79.1*		*75.4*	*71.6*	*77.7*	
At 30 days	127 (6.0%)	6.4%	5.6%	7.9%	0.11	5.0%	4.0%	5.9%	0.43	5.6%	3.4%	7.5%	0.03
*Mean age (years)*	*75.2*	*75.2*	*74.5*	*76.1*		*76.9*	*72.3*	*79.6*	*0.7*	*74.2*	*71.1*	*75.4*	
At 1 year	308 (14.5%)	15.4%	14.2%	17.5%	0.12	15.0%	16.0%	14.2%	0.65	12.2%	9.0%	15.0%	0.03
*Mean age (years)*	*75.4*	*75.3*	*74.* *0*	*77.1*		*74.4*	*73.4*	*75.4*		*76.5*	*74.8*	*77.3*	
**After surgical or percutaneous act**													
At 30 days	83 (6.1%)	6.3%	5.5%	7.8%	0.23	7.7%	3.1%	12.8%	0.02	5.1%	1.6%	9.0%	0.001
*Mean age (years)*	*74.3*	*74.3*	*73.6*	*75.5*		*71.3*	*60.7*	*74.2*		*76.3*	*73.7*	*76.8*	
At 1 year	164 (12.1%)	12.6%	11.8%	14.3%	0.32	13.2%	10.4%	16.3%	0.24	10.5%	6.7%	14.6%	0.01
*Mean age (years)*	*73.1*	*72.8*	*71.7*	*74.8*		*70.7*	*66* *.3*	*73.9*		*75.2*	*75.0*	*75.3*	

* available only for metropolitan France; ** Index hospital stay = the first stay of the year when the patient was hospitalized for MR.

## Data Availability

The data presented in this study are not publicly available.

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
