# Peer review of "Fourteen-Year Temporal Trends in Patients Hospitalized for Mitral Regurgitation: The Increasing Burden of Mitral Valve Prolapse in Men"

_jcm, 2022, doi:10.3390/jcm11123289_

Round 1

Reviewer 1 Report

The paper provides an epidemiologic analysis of trends in mitral regurgitation incidence from 2006-2017 in France and characterizes MR diagnosis and management in the last year (2017) with very detailed tables. Remarkable observations are that there has been increased MR over time in men, while rates have been stable in women; MR related to MV prolapse is increasing over time while other etiologies are stable or decreasing; men are more likely than women to get certain procedures for MR.

Comment:

  1. The paper does not seem motivated by a primary hypothesis. Instead, it reads more like an encyclopedia article. This makes it difficult to read. It would be better if the figures and tables in the paper highlighted a few points that the authors consider important for clinical practice, and they put the remainder of the data in supplemental tables.
  2. The title could also be changed to give the paper greater focus on what the authors consider the most important observations.
  3. Why did the analysis stop at 2017 when it is 2022? While it is understood that it takes some time to get data from registries, I would think that data from 2020 should have been available for this analysis. The problem is that the paper may be missing a trend with respect to percutaneous versus surgical procedures that has occurred between 2018-2020.
  4. There were several formatting issues with tables and figures that should be corrected. For example, the legend is cut off in Figure 1, and the line breaks for the headings in Table 1 split words in the middle.
  5. The English style is good overall but could be improved with further editing.

Reviewer 2 Report

This manuscript describes nationwide dynamics, trends, and epidemiology of the patients diagnosed with mitral regurgitation with history of hospitalization based on the French National Health Data System.

I would like to congratulate the authors of their enormous effort for their nationwide observative study looking at dynamics of the MR patients among French population. The given information would help understand the general idea and context of socio-medical status of the country.

Charlson index is presented to help understand the patients’ physical status, their frailty status, and risk stratification at the time of admission. The authors also presented early outcome as well as 1-year mortality in the subgroup analyses ( Table 2,3). I would like to know that Charlson index would best correspond to which results/mortality (e.g. at 30 during index stay/at 30days/at 1 year, with or without intervention, men/women). According to table 1, more than half of the patients underwent intervention for all the subgroups (MV replacement, MV plasty, percutaneous MV procedure), preoperative risk stratification (e.g. EuroSCORE II) may be more direct indicator. Evaluating the patients by these risk models may also give clues to estimate issues that society may have, for example, whether women are undertreated or underdiagnosed at the time of index hospitalization for heart failure (which have been commented in the text by the authors). If those data are not available, I would recommend to add discussions on these points to give clues for proceeding future studies.

This manuscript provides very important data on socio-economical management of the MR patients who needs medical control or any types of intervention as is commented in the discussion section. It would be more informative that major adverse cerebral and cardiovasclular events (MACCE) during 1-year follow-up be presented in table 2 and 3 in addition to mortalities at any time points for all the subgroups. Preventive medical point of view, understanding the status of adverse outcomes other than mortality may help prioritize the medical approach and treatment choices among patients with primary MR (MP prolapse, rheumatic disease) and secondary MR (ischemic MR, cardiomyopathy etc).

Further English editing is necessary to be published in the distinguished international journal.

Round 2

Reviewer 2 Report

Revised version of the manuscript for the original title with Epidemiology of patients hospitalized for mitral regurgitation in France: Comparisons by sex and etiology in 2017 and 10-year time trends, with newly entitled ’14-year temporal trends in patients hospitalized for mitral regurgitation: the increasing burden of mitral valve prolapse in men’.

The response to the questions were appropriate and well explained with additional tables. However, no negligible amount of data was added in the revised version (data from year 2018 through 2020). This change was very confusing to me that this manuscript is completely new one but not the revision of the original article. The authors should discuss more on the influence of Covid-19 pandemic, which is not so simple.

 I rather enjoyed the original version of the manuscript with the author’s response.

Author Response

We thank the Reviewer for these comments. The data has been extended to the 2018-2020 year after Reviewer 1’s request.

We have completed the discussion regarding Covid19 impact.

“In early 2020, the Covid-19 pandemic significantly affected the French healthcare system, including both in and out-of-hospital care, which led to the cancelling of scheduled non-emergency care on one hand and the saturation of some services, espe-cially those linked to Covid19 management, on the other hand. As observed for several cardiovascular diseases [38, 39], our study found that in 2020, there was a decrease in hospital admissions for MR. A decrease in several cardiac procedures was highlighted in England in early 2020, and mitral procedures were among the most impacted [40]. During a time of major restrictions in services and during peak of pandemic, mitral procedures were recommended only in emergencies or in the most severe patients not controlled by drug therapy [41]. Ambulatory and telemedicine management was en-couraged [42] and could explain part of the decrease in the MR hospital admission in 2020. In addition, some patients were able to delay their consultations or follow-ups on their own, for fear of infection or due to restrictions imposed during the lockdown.

Another hypothesis that could explain the 2020 decrease observed in our study could be the overall decrease in healthcare in France and particularly in general prac-titioners and cardiologists visits that may have limited the detection of MR and led to a decrease in their management. These hypotheses highlights the need for ongoing mon-itoring of this disease because delays in management could have an impact on the epi-demiology of hospitalizations for MR in the coming years, possibly leading to an in-crease in the number of mitral valve repair or replacement or MR complications. Fi-nally, patients with comorbidities or a history of cardiovascular disease were at par-ticular risk of dying from Covid19 in 2020. Thus, we cannot completely rule out that Covid19-related mortality may have been an important competitive cause of death and could have impacted the number of patients with MR [43].”